# An Optimized Small-Scale Rearing System to Support Embryonic Microinjection Protocols for Western Corn Rootworm, *Diabrotica virgifera virgifera*

**DOI:** 10.3390/insects14080683

**Published:** 2023-08-02

**Authors:** Fu-Chyun Chu, Pei-Shan Wu, Sofia Pinzi, Nathaniel Grubbs, Allen Carson Cohen, Marcé D. Lorenzen

**Affiliations:** Department of Entomology and Plant Pathology, North Carolina State University, Raleigh, NC 27695, USA; cchu@fl70inc.com (F.-C.C.); accohen@ncsu.edu (A.C.C.)

**Keywords:** insect rearing, rootworm, molecular biology, function genomics, CRISPR

## Abstract

**Simple Summary:**

The United States Corn Belt consists of large monoculture corn fields. The use of insecticides and insecticidal toxins like Bt to control corn pests such as the western corn rootworm (WCR) has led to increased levels of resistance. While protocols exist for rearing WCR in the laboratory for use in pesticide trials and testing resistance to transgenic crops, they are not optimal for performing genetic engineering. Here we report the development of an optimized rearing system for use in WCR functional genomics research, specifically the development of a system that facilitates the collection of precellular embryos for microinjection as well as gathering large larvae and pupae for downstream phenotypic screening. A quality control system was also established to monitor colony health. This study also provides a model for the development of new rearing systems and the establishment of highly controlled processes for specialized purposes.

**Abstract:**

Western corn rootworm (WCR), a major pest of corn, has been reared in laboratories since the 1960s. While established rearing methods are appropriate for maintaining WCR colonies, they are not optimal for performing germline transformation or CRISPR/Cas9-based genome editing. Here we report the development of an optimized rearing system for use in WCR functional genomics research, specifically the development of a system that facilitates the collection of preblastoderm embryos for microinjection as well as gathering large larvae and pupae for downstream phenotypic screening. Further, transgenic-based experiments require stable and well-defined survival rates and the ability to manipulate insects at every life stage. In our system, the WCR life cycle (egg to adult) takes approximately 42 days, with most individuals eclosing between 41 and 45 days post oviposition. Over the course of one year, our overall survival rate was 67%. We used this data to establish a quality control system for more accurately monitoring colony health. Herein, we also offer detailed descriptions for setting up single-pair crosses and conducting phenotypic screens to identify transgenic progeny. This study provides a model for the development of new rearing systems and the establishment of highly controlled processes for specialized purposes.

## 1. Introduction

The western corn rootworm (WCR), *Diabrotica virgifera virgifera*, is a major agricultural pest that has invaded many corn-producing regions around the world [1]. To make advances in controlling WCR, researchers established laboratory rearing protocols for this species in the mid-1960s [2]. Their rearing protocols were based on methods developed for rearing a closely related species, the southern corn rootworm, *Diabrotica undecimpunctata howardi* [3], and were updated many times during the 1970s and 1980s [4,5,6,7]. Importantly, many of the advances made to date in WCR research owe their success to the availability of lab-reared WCR.

While the aforementioned WCR rearing protocols were effective, it still took at least seven months to rear a single generation because eggs had to be chilled for five months to meet their diapause needs [5]. Therefore, the discovery and establishment of a non-diapausing strain of WCR was a major advancement in WCR rearing, allowing researchers to produce an average of six generations per year, thereby providing a continuous supply of insects for experimental use [8]. This type of strain selection has long been the basis for scientific advancements that can only come from well-focused rearing research [9]. In fact, some of the most significant advances in our understanding of insect biology have emerged from seemingly incremental refinements of rearing processes [9]. An excellent example of this is the research from Huynth et al. [10], who developed a non-diapausing strain of the northern corn rootworm, *Diabrotica barberi*, in conjunction with the development of an improved rearing system.

Most corn rootworm rearing protocols have focused on supplying insects for use in applications such as pesticide trials [11,12,13] or testing resistance to transgenic crops [14,15]. However, the large production scale required to generate the number of insects needed for these experiments (e.g., hundreds or thousands) could withstand inefficiencies in the system since they would have negligible impacts on overall output. Researchers that required fewer insects developed smaller-scale rearing and mating protocols for use in behavioral experiments [16,17], and some tried to optimize rearing protocols for enabling manipulation of specific life stages [18]. Each of these protocols is useful and currently serves as the basis for the laboratory rearing of WCR.

An important requirement and a major challenge for rearing WCR is having high-quality corn roots to serve as larval diet. Most rearing protocols accomplish this by co-localizing the insect with its host (i.e., rearing larvae in soil alongside corn plants) [5,18,19]. Once WCR larvae are ready to pupate, they stop feeding and start searching for a suitable location in the soil for pupation. To accommodate this behavior, many protocols suggest moving late-stage larvae to a soil-only environment [18]. In the absence of satisfactory locations for pupation, larvae will remain in the wandering stage for longer periods of time (up to a few weeks). We have made a preliminary determination that prolonged wandering reduces both fitness and survival. For mating and oviposition, WCR adults are usually kept in cages that accommodate population sizes of 100–1000 individuals.

Established protocols not only pay close attention to the WCR life cycle, as described by Krysan [20], but also take into account WCR characteristics such as oviposition and mating behaviors. The two most common egg-collection methods use either soil dishes [15] or agar plates with cheesecloth [19] for oviposition. Eggs can be collected overnight or for as long as seven days [4,15]. Collected eggs are washed to remove soil and other particulates either by washing the collected eggs on a sieve or by washing them off the cheesecloth into water [15,18]. Both methods have good egg recovery rates.

Functional genomic experiments, including the development and maintenance of transgenic strains [21,22], have certain requirements not currently met by standard WCR rearing protocols. Therefore, we have developed a specialized rearing system that allows manipulation of WCR at different life stages and is efficient for rearing multiple strains on a small scale. Importantly, the rearing system supports other protocols we have established, namely microinjection of precellular WCR embryos for the purpose of germline transformation or CRISPR/Cas9-based genome editing [22]. This report also outlines the basic steps necessary for performing each of the related tasks, such as screening and handling WCR at various life stages.

## 2. Materials and Methods

### 2.1. Source of Insect Strains

The non-diapausing strain of WCR used in this work, and hereafter referred to as a wild type (WT), is a mixed colony composed of insects received from Dr. Wade French (United States Department of Agriculture-Agricultural Research Service-Northern Grain Insects Research Laboratory (USDA-ARS-NGIRL), Brookings, SD, USA), and Crop Characteristics, Inc. (Farmington, MN, USA). The colony was established with ~1000 adults generated from four weekly shipments of ~2000 eggs each. Eggs were reared to adulthood following Dr. Wade French’s protocol [10]. Transgenic WCR were previously developed by Dr. Fu-Chyun Chu [21], following the protocols outlined here and in our previous papers [21,22].

### 2.2. Details of the Rearing Protocol

#### 2.2.1. Rearing Conditions

All insects were maintained under our standard rearing conditions of 26 °C (±1 °C) and 60% humidity (±10%) with a 14:10 light cycle (with 250–330 Lumens of light). Insects were reared on organic Trucker’s Favorite yellow corn (Coor Farm Supply, Smithfield, NC, USA) grown in Scotts^®^ Premium Topsoil (The Scotts Company, Marysville, OH, USA). The new soil was stored at −20 °C for at least 7 days and allowed to warm to room temperature overnight before use.

#### 2.2.2. Preparation of Sprouted Corn

Corn kernels were washed with sodium hypochlorite (1% bleach or 1 mL of bleach in 99 mL of water) and rinsed thoroughly. The kernels were then soaked in tap water overnight and spread on damp tissue paper in a 150 × 15 mm Petri dish for one to two days at 20–22 °C. The kernels were checked after 24 h for evidence of sprouting. Once sprouts reached 10 mm in length, they were placed in a 950 mL plastic box (see Appendix A for a full list of containers and sources), secured with a lid, and stored at 4 °C until needed. One box of sprouted corn was enough to create 40–50 primary rearing containers (475 mL plastic cups) or 20–25 secondary rearing containers (950 mL plastic boxes).

#### 2.2.3. Single-Pair and Colony-Level Egg Laying

The agar-based oviposition plates used in this protocol consisted of a 1% solution of melted agar (Drosophila agar, Type II, Apex) in 100 × 15 mm Petri dishes, poured to a depth of 10 mm. Once cooled, a layer of filter paper was placed on the surface of the solidified agar and covered with four layers of cheesecloth.

For a small egg collection, such as from a single-pair mating, WCR adult diet (western corn rootworm w/o pollen substitute, Frontier Scientific, Newark, DE, USA) was added to a 35 × 10 mm plastic Petri dish, placed on an oviposition plate, and the plate covered with a form-fitting 180 mL plastic container (see Appendix A) having an equivalent opening (100 × 15 mm, similar to a Petri dish). To ventilate, a steel pin was used to make approximately 30 holes in the “top” (=bottom of the 180 mL plastic container). Two to twelve adults were added to each single-pair chamber, and a fresh adult diet was provided every two to three days to avoid mold growth. These chambers were used for both overnight and weekly egg collection but were replaced on a weekly basis to avoid agar desiccation.

For colony-level egg collection, an oviposition plate was placed inside a 30 cm^3^ cage (BugDorm, MegaView Science, Taiwan) containing 500–1000 adults. The cage also held a water supply: a 300 mL flask of water covered with a cotton ball that held a 15.2 × 1 cm cotton roll in place. A 100 × 15 mm Petri dish with adult diet was placed in the cage, and diet was added as the original diet was depleted or became too dry. The oviposition plate was covered with a tinfoil tent [22]. Overnight oviposition is recommended, but the system is also suitable for collecting embryos over a longer span of time, up to 5 days.

#### 2.2.4. Egg Collection and Incubation

A pair of forceps was used to remove the cheesecloth, one layer at a time, and to gently swirl each layer successively in a 500 mL beaker of water to dislodge the eggs. A wide-bore 5.8 mL transfer pipette was used to aspirate the eggs, while minimizing the amount of water carry-over, and place them on filter paper or into soil. If the water also had lots of debris or unwanted material, the eggs were transferred into a new beaker of water for another wash; this was repeated up to three times. To maintain colony size, 300–500 eggs were directly transferred into each of two to three 30 mL cups with topsoil and a lid (Figure 1) and held for seven days in an insect-rearing incubator set at 26 °C, 60% humidity, and a 14:10 light cycle. Two to three cups of eggs (600–1000 total) per week were adequate for maintaining a colony of 500–1000 active adults. A single colony of this size can provide more than 10,000 eggs per week.

For other purposes, where eggs or first-instar larvae are needed, the desired number of eggs were placed on filter paper using a transfer pipette and incubated on an agar plate at 26 °C. This method was used to keep the eggs until they hatched. We observed occasional mold growth on the filter paper or dead eggs during incubation, but this did not appear to affect hatch rates or other apparent fitness parameters.

#### 2.2.5. Preparing Primary Rearing Containers

After incubating the collected embryos for seven days, the entire contents of each 30 mL cup were transferred to a primary rearing container (475 mL container) containing sprouted corn (Figure 2), and then extra soil was added to fill 60% of the container. To ensure proper containment of insects, each primary rearing container had a lid. Each lid was perforated with small holes using a dressmaker/sewing pin (0.6 mm diameter) for air exchange (Figure 2). Water was added as needed to wet but not oversaturate the soil. For larval rearing conditions, too much moisture increased mold and mite issues; however, too little resulted in dry roots or larval death. To test soil moisture levels, samples of soil were squeezed between the fingertips; ideally, there was enough moisture to hold the soil together but not so much that water squeezed out. However, the best conditions differed depending on the temperature and humidity of the rearing room. Typically, eggs hatch 2–4 days later, and larvae start feeding on the corn roots. Despite their small size, these containers held enough corn roots to support larvae until mid-to-late second instar (~1 week at standard rearing conditions).

#### 2.2.6. Preparing Secondary and Tertiary Rearing Containers

Secondary rearing containers (950 mL) were prepared at least three days before use by adding sprouted corn covered with soil to only one side of the box (Figure 3A). If the rearing containers were prepared more than three days prior to use, they were stored at 4 °C. Fourteen days after egg-laying (AEL), everything from the primary rearing container was transferred to the empty half of the secondary rearing container (Figure 3B). If any larvae remained in the primary rearing container after removing the soil and corn, a small brush was used to carefully transfer the larvae to the secondary rearing container. More soil was added to fill 60–70% of the box, and the box was kept at standard rearing conditions.

Approximately two weeks AEL (25–28 days), secondary rearing containers were opened, and larvae and pupae were sorted and transferred to new containers. An indication that the insects were healthy was if most were late third-instar larvae, pre-pupae, or pupae. Third-instar larvae still need food, so these were moved to a tertiary rearing container (950 mL) (Figure 4) containing evenly spread sprouted corn covered with a layer of soil. Only 100–150 large larvae were transferred to each tertiary rearing container to ensure suitable support until they eclosed.

Importantly, late third-instar larvae are relatively large and strong enough for handling; therefore, this is the best age for phenotypic screening. It is also the best stage for microinjecting double-stranded RNA since these larvae are more tolerant of damage that can occur during the injection process. Moreover, it only takes an additional 10–14 days to reach adulthood, which makes it easier to track individuals or groups of insects for data collection. However, pre-pupae and pupae are not good for microinjection and need to be handled with great care. Too much force from a paint brush or forceps can lead to deformed adults or unsuccessful eclosion.

#### 2.2.7. Preparing Pupal Rearing Containers

Pre-pupae and pupae were placed into a 950 mL plastic container with soil but no corn (Figure 4). Each container can hold 150–200 insects. All insects were covered with soil to avoid desiccation. If individual pupae needed to be kept separately, they were isolated in 30 mL cups with soil, like those used for embryos, until eclosion. For example, pupae were often sorted by sex and held in two separate cups, male or female, to ensure virgin matings.

#### 2.2.8. Adult Collection for Colony- or Single-Pair Mating

At approximately 38 days AEL, adults would start to eclose. Since this occurred in the final larval and pupal rearing containers (Figure 4), these were checked every day, and newly eclosed adults were moved to colony cages or placed in mating chambers for use in single-pair crosses (described above). Rearing containers continued to be checked daily until no adults were found for three consecutive days. In cases where recovering every individual was of paramount importance, we briefly removed soil and plants from each container and carefully searched them for remaining insects. Any larvae or pupae discovered during this search were transferred to new containers and allowed to complete their development. Although these measures were labor-intensive and not suited for mass-rearing, they were important in our studies of microinjection, where each individual was highly valuable towards the assessment of our efforts.

### 2.3. Development Time, Eclosion Efficiency after Handling, and Quality Control

Following this protocol, we first collected developmental and survivorship data to determine the rate of larval development and the total developmental time from oviposition to adulthood. Then we tested whether our handling procedures affected the survival rate. After we recovered and sorted insects from secondary rearing containers, we used the larvae and pupae to establish separate rearing containers based on the rearing protocol outlined above and quantified the development of individuals to adulthood. An individual’s size as well as degree of movement and feeding were used as indicators to determine developmental stage, with third-instar larvae being the largest individuals still moving and feeding and pre-pupae being those that had already built a soil cocoon or were otherwise showing little movement. We followed the rearing system for 19 weeks to collect larval and eclosion numbers and calculate the survival rate. Larval and prepupal/pupal survival rates were compared to determine which life stage is most sensitive to handling. A full year’s worth of recorded survival rates was used to set up quality control for monitoring the rearing system. Because survival from third-instar larva to adult is the most sensitive part of the system, our quality control was calibrated by the survival rate after the third-instar larval stage. We used JMP^®^ (SAS Institute Inc., Cary, NC, USA) to calculate and graph the quality control system and to calculate the mean survival rate ($\overset{-}{x}$) and standard deviation (σ). The mean was designated as the standard for the system. The upper control limit (UCL) was calculated as ($\overset{-}{x}$ + 3σ) and the lower control limit (LCL) was calculated as ($\overset{-}{x}$ − 3σ). 

### 2.4. Fluorescent Light Screening

Rearing containers were opened, and insects were sorted approximately 4 weeks AEL, since phenotypic screening was expected to be clearer in large larvae, pre-pupae, and/or pupae. During the screening process, larvae were held at 4 °C to slow activity. The whole screening process for insects in each secondary container took about 20–30 min. We separated recovered individuals into groups of larvae, pre-pupae, or pupae for convenience of screening. Either separated or mixed stages were placed into separate containers and reared to adulthood following the rearing protocol (described above), and survival rates were calculated.

### 2.5. Single Pupae Handling and Survival Rate

We also separated individual pupae for sexing, as described previously [20], and for monitoring the eclosion rate. Handling and screening pupae can sometimes damage them. However, handling pupae is unavoidable because WCR larval development can vary greatly, despite our care to use uniform conditions. Therefore, rearing containers include a range of life stages, from as early in development as second-instar larvae to as late as pupal stage, even if the eggs were laid at the same time. This phenomenon was described as “straggling” for gypsy moth rearing [23,24] and by Cohen [9] for several other species of insects under controlled rearing conditions.

Moreover, for some experiments, such as RNAi or germline transformation, it is likely that individuals need to be screened for phenotypes and separated for monitoring at the pupal stage. As a result, it was important to assess the impact of handling and screening pupae to determine if this method could be a standard part of our WCR rearing protocol. Pupae were removed while sorting each container, and they were moved around with pointed featherweight forceps (Catalog #4748, BioQuip Products, Compton, CA, USA), sexed under a dissecting microscope (Leica, Wetzlar, Germany), and individually placed into a 30 mL cup (one pupa per cup). We then monitored the eclosion rate to determine if handling damaged the pupae and re-sexed them as adults to confirm the accuracy of pupal sexing.

### 2.6. Testing Single-Pair Mating Scheme

Finally, we tested our single-pair mating and egg-laying methods. A single pair of WCR adults were placed into each of the ten mating chambers (see Section 2.2.3), and cheesecloth was changed daily to monitor and quantify each egg-laid. If the male died before the female started laying eggs, a new male was added to the chamber. If the male died after the female started laying eggs, it was simply discarded (no new male was added). Eggs were counted every day, and female survivorship was recorded.

To determine if mating occurred during the 24 h window between the daily collection of WCR adults, a transgenic strain was used to track the parentage. The transgenic strain [21] carries a fluorescent marker gene (DsRed), driven by an eye-specific promoter (3 × P3) [25]. Fifteen outcrosses were established using one transgenic male and two to three wild-type females per container (single-pair chamber). Four reciprocal crosses (one transgenic female and one wild-type male) were established as controls. Third-instar F_1_ larvae were screened for marker gene expression using a Leica M165 FC fluorescence stereomicroscope (Leica Microsystems Inc., Wetzlar, Germany) outfitted with a DsRed filter set (excitation filter: 510–560 nm, emission filter: 590–650 nm). To determine if wild-type females had previously mated with wild-type males, we calculated the ratio of transgenic F_1_ to total F_1_ recovered from each outcross. Moreover, if a wild-type parent had “pre-mated” in the rearing container (WT × WT), the expectation is that progeny from the earliest egg lays would lack DsRed expression.

## 3. Results

### 3.1. Developmental Rate

To determine if this system (Figure 5) meets the requirements necessary for conducting germline transformation, CRISPR/Cas9-based genome editing, or other functional genomics assays in a modest-sized molecular genetics lab, we first assessed the impact of the system on developmental rate. This was accomplished by following a cohort of wild-type WCRs from egg to adult. The cohort consisted of individuals that emerged from six oviposition plates, each plate having between 300 and 500 embryos. After four weeks (25–28 days AEL), a total of 1300 individuals were recovered. The majority were third-instar larvae, while only 13.8% (180 out of 1300) were pre-pupae/pupae. All insects were placed into new containers (see Section 2.2.6) and monitored until eclosion. Of these, 1097 were successfully eclosed (84.4%). The average developmental time from egg to adult was 42.6 days. More than 80% of the adults eclosed between 41 and 45 days (Figure 6), which is similar to other WCR rearing protocols [5], both for overall survival rates from larva to adult as well as development time. Thus, we conclude that the short, high peak seen for adult emergence indicates that the system was stable and that the insects developed at similar rates.

### 3.2. Survival Rate

To obtain additional rearing data over a longer period of time, we monitored 25 smaller cohorts for 19 weeks (see Table 1). Interestingly, larval survival to adulthood was much higher (82.1%) than the survival of pre-pupae/pupae to adulthood (56.0%). However, the average of the two (71.6%) was comparable to survival rates from containers having mixed life stages (73.9%), indicating that both methods work equally well.

### 3.3. Quality Control System

A quality control (QC) system can be used as an indicator for how well a rearing system is working [9,26]. Using the survival data, we determined that, in our system, a mean survival rate of 66.3% could be used as the standard for a healthy colony. This also set upper/lower control limits (UCL/LCL), which could be used as indicators for when the system was out of control and in need of changes to help the overall health of the colony. When we analyzed the data (Appendix A) according to quality control standards, we identified one timepoint that did not pass quality control (Figure 7) (i.e., the survival rate was lower than LCL). Since the rest of the timepoints clearly passed our QC standards, we concluded that the system was stable and only occasionally encountered situations that resulted in reduced survival rates. These data also suggest that if a low-survival situation fails to recover within two weeks, the system is no longer stable and requires interventions or adjustments to the rearing process. Such repairs constitute an operant process control system [9]. It should be noted that our survival data (Figure 7) indicates the rearing system was more stable during the second half of the year. This suggests that when setting up a new rearing system, it might take a few generations (in this case, three generations) to stabilize the rearing system. However, the data can be helpful for generating a robust quality control system.

### 3.4. Impact of Fluorescent Light Screening

To determine if the fluorescent light or cold temperatures (4 °C) could potentially harm the insects, we used the wild-type WCR strain to assess the impact screening has on individuals of various life stages. Of the 480 larvae screened, 438 (91.3%) successfully emerged as adults (Table 2). Pupae had a lower survival rate (82.9%), while pre-pupae had the lowest (68.4%). Importantly, a group containing both pre-pupae and pupae had a slightly better (79.7%) survival rate. Taken together, it does not appear the screening method or cold temperatures caused significant harm.

### 3.5. Handling and Sexing Pupae

In our previous tests, handling pupae resulted in higher mortality than handling larvae. However, it is almost impossible to avoid working with pupae. Indeed, many applications might even require screening pupae. Therefore, we designed and tested an experimental protocol for working with pupae and/or separating them for individual monitoring. A total of 178 pupae were placed into individual cups, and 141 (79.2%) successfully emerged as adults. Given the similar survival rates (compare Table 1 and Table 2), this suggests that additional handling does not cause excessive harm. Therefore, we concluded that it is acceptable to separate individual pupae if the need arises. In addition, only one individual was incorrectly separated based on sex.

### 3.6. Single-Pair Crosses

Because techniques like germline transformation require injectees to be individually outcrossed (single-pair crosses) as part of the experimental design, we developed and tested a protocol for establishing single-pair crosses and obtaining embryos shortly after oviposition. Eight of ten crosses had successful matings, while two had issues. Specifically, two males died before the female started to lay eggs, so new males had to be added. Pair #10 had the longest oviposition delay (~40 days), and the average oviposition delay across all pairs was 19.3 days. After the initial oviposition, there was an average gap of 4.6 days before the next clutch was laid (Table 3). In this system, the females laid an average of eight egg clutches during their lifetime, with the most productive female having 17 clutches and the least productive laying only three (Table 3). The average clutch size was 54.4 eggs per female, while the overall lifetime average number of eggs per female was 414. The shortest-lived female survived for 31 days, while the longest lived for 132 days, with an overall average lifespan of 78.4 days (Table 3).

Since adults were collected at intervals of approximately 24 h, we wanted to determine if mating occurred prior to their collection (WT females × WT males). In this test of the single-pair mating scheme, 15 transgenic males were individually outcrossed with wild-type mates. If pre-mating had occurred, wild-type females would produce non-transgenic offspring for at least the first few oviposition events. Moreover, if the single-pair mating scheme was not efficient or favorable for mating, no F_1_ offspring would be recovered. Importantly, all 15 crosses produced transgenic offspring, even during the first week of oviposition (Appendix A). This demonstrates that all 15 transgenic males successfully mated. However, because the single-pair crosses were not truly “single pairs”, but rather each male had multiple females per chamber, we cannot rule out the possibility that some females may not have mated with the transgenic male.

To gain greater insight, we compared these results with control outcrosses, true single-pair crosses having one transgenic female and one wild-type male per chamber. The results show that 28.4% of the F_1_ offspring from the control crosses were DsRed-positive, similar to the 23% DsRed-positive F_1_ offspring obtained from the transgenic male outcrosses (Appendix A). The ratio of transgenic to wild-type F_1_ progeny found in the control outcrosses ranged from 16% to 40%, while two of the experimental outcrosses (#8 and #11) had ratios below 10%, suggesting that pre-mating (mating of wild-type individuals prior to collection) may have occurred in these two crosses. However, since every outcross provides transgenic offspring in the first week of successful oviposition, pre-mating does not appear to cause an issue in this rearing system.

Here we have demonstrated that our modified rearing system allows manipulation of WCR at different life stages and is efficient for rearing multiple strains on a small scale. Importantly, this rearing system was created to support other protocols we have established, namely microinjection of precellular WCR embryos for the purpose of germline transformation or CRISPR/Cas9-based genome editing [22]. We have also outlined the basic steps necessary for performing each of the related tasks, such as screening and handling WCR at various life stages. In addition, we have shown that our method for carrying out single-pair crosses is efficient for crossing and collecting eggs from individual beetles. Moreover, this system can facilitate the development of additional functional genomics tools, such as transposon-based genome-wide mutagenesis, for future research in WCR.

## 4. Discussion

The goal of this research was to develop a small-scale, well-controlled rearing system for WCR to facilitate functional genomic studies, such as germline transformation, that require manipulating different life stages, screening for phenotypes of interest, and producing high-quality adults for establishing or maintaining transgenic or mutant strains. The results presented herein indicate that our rearing system achieved stable survival and developmental rates and provided sufficient quality control. Each test demonstrated that the manipulation processes and protocols are efficient for obtaining reliable results. Moreover, the single-pair crossing scheme outlined above resulted in successful mating and oviposition and fit easily into the overall system.

Importantly, since this rearing system was specifically designed for raising transgenic and mutant WCR, containers not only had to support larval growth but also provide effective containment. Specifically, for each transgenic or mutant WCR line to remain pure, individuals had to be secured in separate containers to avoid the possibility of cross-contamination. Moreover, this also helped ensure that genetically modified individuals could not escape the confines of the lab. Therefore, each container was covered with a lid that restrained WCR movement while permitting air exchange. While the lids reduced fluctuations in temperature and humidity, the environment inside the rearing containers remained a major concern. The problem stems from the fact that WCR larvae require corn plants for good larval growth. However, the lids limit the available space for plant growth, negatively impacting plant health and, in turn, larval health. We had issues with this while developing our rearing protocol. Specifically, we discovered that corn leaves rot very quickly, rapidly increasing the humidity inside the container, thus causing WCR death, mold growth, and mite outbreaks. However, once the appropriate number of plants per container was determined and the size of aeration openings optimized, the occurrence of such problems was greatly reduced. Specifically, the occurrence of mites and mold was reduced ~160-fold, from ~40 rearing containers per week to less than 1 per month.

Along with reducing mold and mites, stabilizing the larval rearing containers also had an impact on the developmental rate. The time required from egg to adult dropped to 41–45 days, and the prior issues with straggling were largely averted, suggesting that the insects were healthier. Adult females still needed another 10–15 days before they started laying eggs; therefore, the total generation time in our system was about 55–60 days. This is similar to what others have reported [5,15,27]. It should also be noted that the insects in the aforementioned studies were handled by sorting out larvae or pupae, thus making our rearing process comparable to previous methods. While the work presented herein was carried out by three different workers (a graduate student, a lab assistant, and an undergraduate research assistant), the data were not considered replicates but rather a way to demonstrate the accessibility of the system (i.e., any level of worker can be trained to use or handle this system). This is a crucial point, because any source of variation is a basis for an unacceptable loss of control over the entire rearing process, and operator error is one of the greatest sources of failure in rearing systems [9].

Our survival and eclosion rates are also similar to those found in other studies [5] and are suitable for maintaining a healthy colony and avoiding concerns about not having enough adults to provide adequate numbers of eggs. Each of our colonies had 500–1000 adults and provided over 10,000 eggs per week. Moreover, analysis of the rearing data using the quality control standard outlined above indicated that our colony is relatively stable. We also found that this rearing system could endure occasional low survival rates. This is primarily because we consistently worked to maintain the colony by collecting new eggs every week instead of operating on a generation-to-generation timescale, as is done with other species. As a result, one week of high losses has little impact on the overall system. On the other hand, if low survival rates are a continual problem, this would indicate a fundamental change in the environment or a breakdown in the system. If this occurs, immediate action is required to find out where the problem lies.

We were not able to fully assess hatch rates for colony-level rearing in our process control system because eggs were not counted (i.e., numbers were estimated) and they were placed directly into the soil, making it impossible to monitor their development. We did, however, check hatch rates by placing eggs on moist filter paper and monitoring them over time, but this is not the same as hatching in the soil. The major difference between incubating eggs on moist filter paper vs. in soil is the increased mold and bacterial growth when using filter paper. Although the presence of mold and bacteria lacked an overt impact on hatching, the fact that they change the environment means that these microbes may have less obvious impacts on WCR health and fecundity.

As mentioned above, the main reason for developing this small-scale rearing system was to facilitate functional genomic studies in WCR. Since the first of these studies was aimed at generating transgenic WCR marked with a fluorescent protein gene [21], we were concerned about the impact of exposing WCR larvae to the intense light waves used in fluorescence microscopy. Interestingly, when larvae or pupae were exposed to fluorescent light, they reacted negatively, obviously attempting to avoid the light (negative phototropism). However, the cold temperature of placing them on ice (4 °C) slowed their movement enough to enable successful screening. While it was not immediately clear if exposure to fluorescent light or cold was harmful to WCR, our data indicate that both larvae and pupae survive exposure to these sufficiently well.

Another typical step in the process of establishing transgenic strains is setting up single-pair crosses, but this mating scheme has received sparse attention for WCR. One notable exception is Hill et al. [17], who reported data from 11 single-pair crosses (see Table 3). Our results were comparable to Hill’s in most categories, even though Hill’s methods involved feeding adults with natural food instead of an artificial diet, and their experimental insects were field-collected larvae, not lab-raised ones. Another group, Branson and Johnson [16], reported finding twice as many eggs per female as we did and observing greater female longevity (see Table 3). Other groups used a diapausing strain of WCR from the wild or only kept the insects in the lab for about 10 generations. This makes comparison to our work difficult because our WCR are from a non-diapausing strain that has been kept in the lab for more than 200 generations. However, these data do suggest that our rearing system, and in particular, our single-pair crossing scheme, works to produce and maintain adults and embryos of sufficient health and fecundity for functional genomics research.

In our system, adults were collected daily. However, this leaves a 24 h period where newly eclosed adults could have mated. Importantly, previous reports suggest that WCR are more likely to mate in the early morning and evening [28]. Therefore, since we usually collect adults in the late morning (10–12 a.m.), it is possible that some beetles mated before we collected them. While male WCR can mate multiple times, females usually only mate once during their lifetime, though some have been observed to mate twice [17]. Branson and co-workers [29] demonstrated this experimentally, showing that, after being exposed first to sterile males, then to fertile males, only seven out of thirty WCR females produced viable eggs. Since these reports indicate that it is rare for WCR females to have successful second matings and that their eggs might be fertilized by sperm from the first mating, we used transgenic (DsRed-tagged) males to determine if the collected females were virgins.

As reported above, all outcrosses between wild-type females and transgenic males produced transgenic offspring. While this demonstrates that females did indeed successfully mate with transgenic males, the fact that each transgenic male was mated to three females means that it is possible that some females may have pre-mated with wild-type males, but the percentage seems to be very low. Another point to consider is that this test used several transgenic strains, some of which were very weak expressers, which could easily explain why some crosses had low percentages of transgenic offspring. In fact, it was common to see low percentages of transgenic progeny from these strains even when performing self-crosses (transgenic male x transgenic female). Moreover, this conclusion is supported by the control crosses (transgenic females mated with wild-type males), which also showed low percentages of transgenic progeny.

## 5. Conclusions

The rearing system reported here was designed specifically to support functional genomic experiments. It is geared towards small-scale studies requiring multiple strains or treatments, with the added requirement of having to fit within the confines of an average-sized molecular biology laboratory. This system also allows manipulation of WCR at different life stages, which is necessary when screening for genetic traits such as eye color or fluorescence, which are frequently used as markers when establishing transgenic or knock-out strains [21,30,31]. Importantly, it also addresses containment since reducing the risk of WCR escaping the confines of their rearing containers is of vital importance to both maintaining pure breeding strains and ensuring the insects never breach the confines of the laboratory. This is especially important when working with pest species like WCR or any insect that has been genetically modified. In addition, we have shown that our modified single-pair crossing scheme can be an efficient method for mating individual beetles and collecting their embryos. A future objective is to use this system to generate mutations in WCR genes that are orthologous to ones known to be involved in Bt-resistance in lepidopteran species. Such site-specific mutations should allow researchers to gain a better understanding of WCR genetics, genomics, and biology. The availability of small-scale rearing protocols tailored specifically for functional genomic studies of WCR will hopefully expand the use of the newest genetic tools (e.g., CRISPR/Cas9) in the study of this important pest, as well as serve as a model for the development of specialized rearing systems for other species.

## Figures and Tables

**Figure 1 insects-14-00683-f001:**
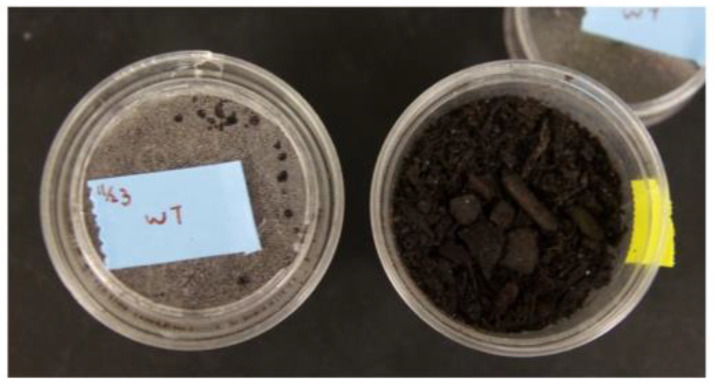
Egg rearing containers. Eggs are placed in the middle of a 30 mL cup and covered with soil.

**Figure 2 insects-14-00683-f002:**
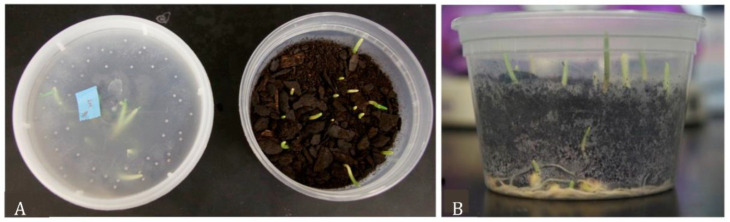
Primary rearing containers. (**A**) Each 475 mL container possesses soil and newly sprouted corn; lids have holes to allow air exchange. (**B**) Minimum level of root growth required for larvae to feed.

**Figure 3 insects-14-00683-f003:**
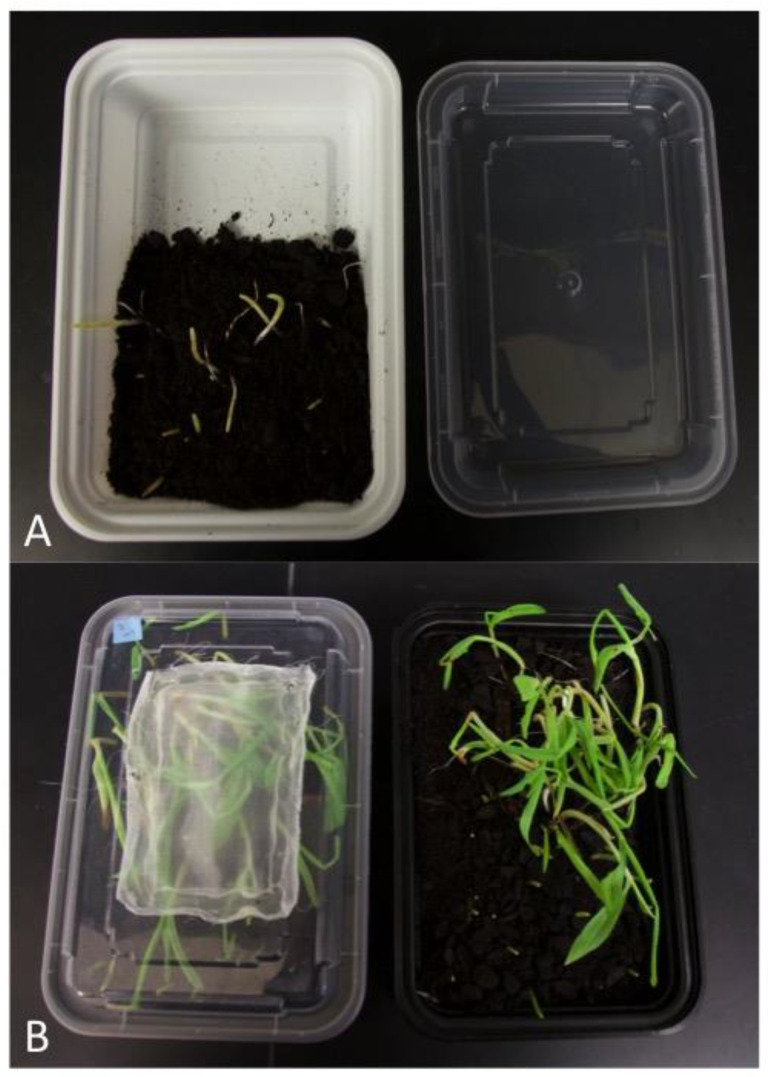
Secondary rearing containers. (**A**) The half box set up: the empty half of a 475 mL container will be used to hold the contents of a primary rearing container following transfer. The intact lid should be used to cover the growing corn prior to adding insects. (**B**) After adding the contents from a primary rearing container, use a lid with a fine screen for air exchange.

**Figure 4 insects-14-00683-f004:**
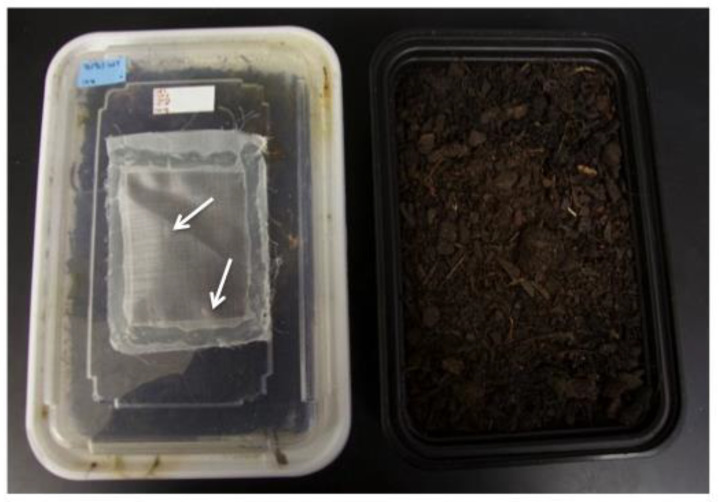
Tertiary or pupal rearing containers (adult collection containers). Adult collection containers can be either tertiary rearing containers (with plants) or pupal rearing containers (without plants). Most adults exit the soil and can be seen through the lid (arrows point to newly eclosed adults). Once pupae start eclosing into adults, all plants can be cut back and removed to increase visibility, thereby aiding the collection of adults.

**Figure 5 insects-14-00683-f005:**
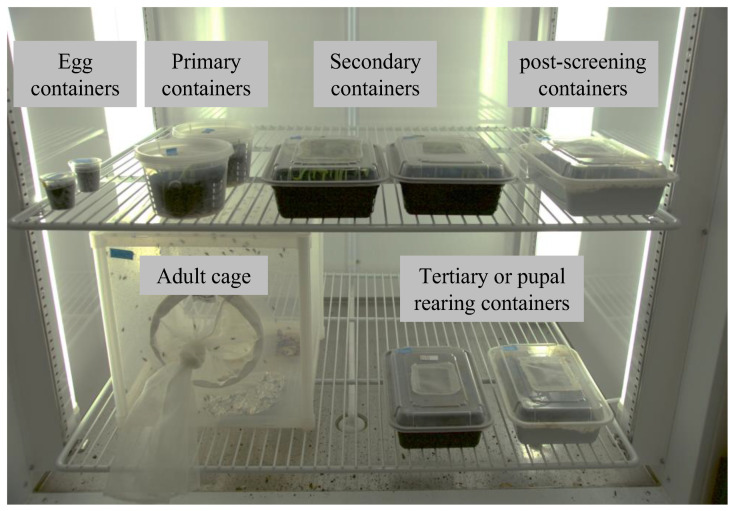
Containers required for a single WCR colony. The minimum space and assortment of containers required per colony using this rearing system. Upper level, left to right: two egg cups, two primary rearing containers, two secondary rearing containers, and one post-screening container. Lower level, left to right: adult cage and two adult collection containers.

**Figure 6 insects-14-00683-f006:**
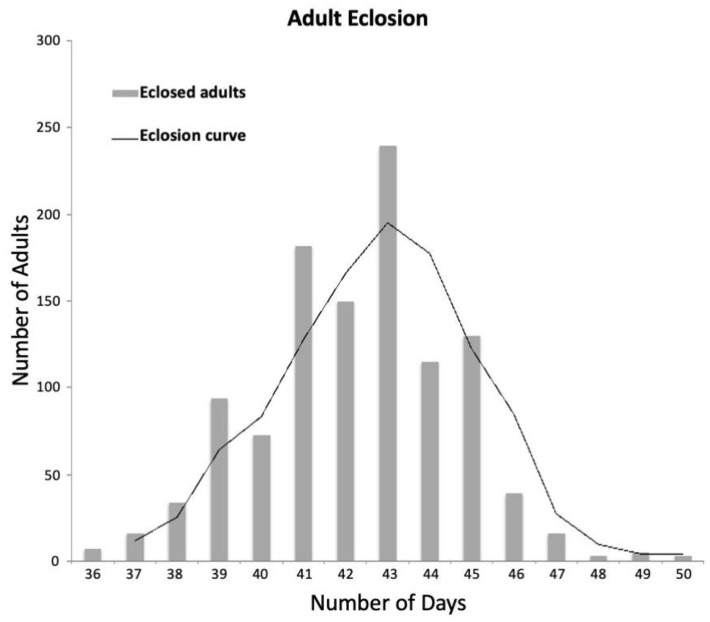
WCR development time. Each bar indicates how many insects had eclosed by each of the given total days of development time (days). The average is 43 days.

**Figure 7 insects-14-00683-f007:**
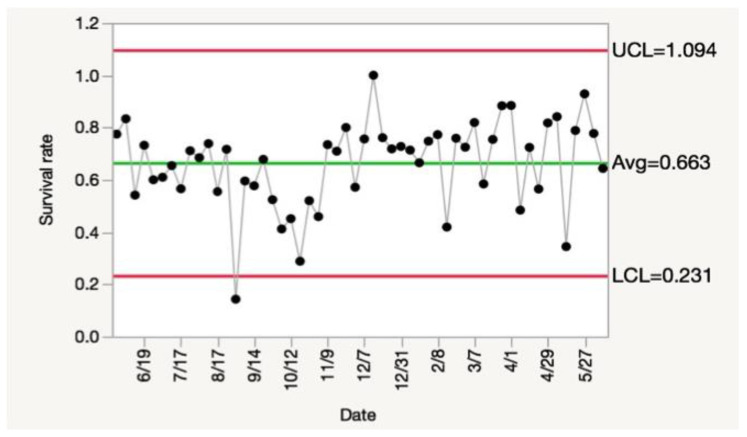
Quality control assessment. This quality control system is based on the survival rates (black dots) for WCR (see Appendix A). Survival rates between the upper red line (UCL) and lower red line (LCL) are within quality control standards, but those outside would be considered to be out of control. UCL (Upper control limit) = x + 3σ (x = mean, σ = standard deviation). LCL (Lower control limit) = x − 3σ.

**Table 1 insects-14-00683-t001:** Survival rate among different life stages of WT WCR.

Stage	# of Insects	# of Adults	Survival Rate
Larvae	3260	2675	82.06%
Pre/pupae ^1^	2184	1223	56.00%
Mix ^2^	2146	1586	73.90%
Total	7590	5484	72.25%

^1^ Pre-pupae and pupae were grouped together since they no longer require food. ^2^ Larvae, pre-pupae, and pupae in the same container.

**Table 2 insects-14-00683-t002:** Effect of fluorescent marker (CFP) screening on adult eclosion rate.

Life Stage	# of Insects	# of Adults	Eclosion Rate
Larvae	480	438	91.25%
Pre-pupae	2630	1798	68.37%
Pupae	4766	3950	82.88%
Pre/pupae	818	652	79.71%

**Table 3 insects-14-00683-t003:** Female fitness and fecundity in single-pair crosses.

	This Work		1973 *		1972 ^+^	
	Range	Mean	Range	Mean	Range	Mean
Female longevity (days)	31–132	78.4 ± 32.6	6–163	67.7 ± 30.6	19–126	94.8 ± 12.5
No. eggs oviposited/female	83–1070	414 ± 290.3	237–912	593.l ± 231.3	85–1913	1023 ± 240
No. clutches/female	3–17	8.1 ± 5	3–20	11.3 ± 5.4		
Avg no. eggs/clutch/female	1–128	54.42 ± 25.7	39–79	56.9 ± 14.2		
Avg days between ovarian cycles	1–11	4.63 ± 2.3	4.2–6.0	4.9 ± 0.4		
Preoviposition period (days)	13–40	19.3 ± 8	11–19	15.3 ± 2.0		

* Hill (1975) [17]; ^+^ Branson and Johnson (1973) [16].

## Data Availability

All data are contained within the article and Appendix A.

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
