# Peer review of "An Optimized Small-Scale Rearing System to Support Embryonic Microinjection Protocols for Western Corn Rootworm, Diabrotica virgifera virgifera"

_insects, 2023, doi:10.3390/insects14080683_

Round 1
Reviewer 1 Report
This is a thorough report of rearing procedures for western corn rootworm, and it offers a valuable background for future genetic studies on this species. The text is well written and sufficiently detailed as to allow replication of the procedures by other workers, which is a key intent of the authors. I found no errors or omissions in the methodology, and have very minor suggestions for corrections to the text. Specifically:
line 38: "To" instead of "In order to"
line 55: incomplete italics on "barberi"
line 230: change location of apostophe "An individual's..."
line 304: " a total ... was recovered."
see notes above
Author Response
We have made each of the suggested changes.
Specifically,
line 38: is now "To" instead of "In order to"
line 55: we have completed the italics on "barberi" (thanks for catching that)
line 230: we changed the location of the apostophe in "An individual's..."
and changed line 304: to " a total ... was recovered."
Reviewer 2 Report
The rearing protocol developed by the authors for procuring embryos for micro injection and getting larvae and pupae for screening does helps to get these insects for genetic manipulation at different stages.
Few minor things which needs to be added to the manuscript:
1. What type of soil and soil preparation (autoclaved or not)
2. Do put axis titles on the figure 6.
Author Response
- We have added information to the text about the type of soil and soil preparation.
- We put axis titles on the figure 6 (thanks for catching that).
Reviewer 3 Report
The paper is within the scope of the journal and is of certain interest to its readers. Diabrotica is a dangerous pest with high invasive potential that keeps on spreading all over the world, posing a threat to corn production. The manuscript is well-written. The introduction provides an in-depth dissection of the problem and covers both historical retrospective and future prospective of the study. The Methods are smart, detailed and clear, this part of reading is particularly enjoyable. It is easy to imagine oneself reproducing all the procedures as they are described, especially given that the pictures of all stages of rearing are provided (but see below). This is very important for the studies in this field. The paper deserves publication after correction of some minor flaws listed below.
Illustrations are good enough by design by their quality is a bit lame.
L12: has led to populations – consider revision
L18: more accurately monitoring colony health – reconsider the use of adeverb and pretext
L18 & L32: “model” is mentioned only in the abstract/summary and can’t be find in the main body of the manuscript
L78-82: references are lacking
L85-94: the explanation of the research achievements can be contracted in Introduction and expanded in Conclusion
L98-104: the amount of insects used for lab colony establishment is a desirable item
L157-158: the desired number of eggs – please give the thresholds (10-100 etc)
L160: mold growth – where exactly?
L168: small holes – diameter is welcome
Ibidem: water – can some approximate amount (volume ratio) be given?
L82: days-AEL – hyphen seems to be odd here
L206: from desiccating = from desiccation
L225: data to determine – what kind of data is meant?
L233: still looked like larvae – sounds like there are insects which are not larvae but look like larvae :) please rephrase
L237-238: to set up a quality control system for monitoring the rearing system = to set up a quality control for monitoring the rearing system (avoid excessing word repetition, “control” in this case)
L238-238 from the third-instar larval stage to the adult stage = from the third-instar larva to the adult
L254: rates calculated = rates were calculated
L277-278: “pairs vs chambers” doesn’t give a clear comprehension if there were ten pairs per chamber or ten pairs per ten chambers
L279: quantify egg-laying – consider “quantify the egg-laying”
L281: he was – consider “it was”
L284: track parentage – consider “track the parentage”
L300: lab we – consider comma usage
L308-310: please build up this conclusion upon some previously published data
L326: you may want to mention the overall period of observation spanned by the mean survival rate
Table 2 – the CFP type is indicated in the heading and is homogenous for all variants, the first (from the left) column should be removed
L473: what does the sentence in the parentheses mean?
L482: do you consider the work performed by workers of three different levels as independent replicates and compare the data between to access reproducibility?
L556-567: this last paragraph may be headed as “Conclusion” as a lot of information herein is a repetition of ideas already given in Discussion.
Author Response
We have made each of the suggested changes.
Specifically,
We have worked hard to improve the resolution of our graphical abstract and hope it is now of sufficient quality. Apologies for originally submitting one of such poor quality.
L12: has led to populations – has been changed to “has led to increased levels of resistance”.
L18: more accurately monitoring colony health – We have deleted “more accurately”.
L18 & L32: “model” is mentioned only in the abstract/summary and can’t be find in the main body of the manuscript. We have added “model” to the intro and conclusions to make clear that the rearing system serves as a “model” for how to go about creating similar rearing systems for other species.
L78-82: references are lacking – We have inserted citations (thanks for catching this).
L85-94: the explanation of the research achievements can be contracted in Introduction and expanded in Conclusion – Thanks for the suggestion! We have moved and revised the relevant paragraphs.
L98-104: the amount of insects used for lab colony establishment is a desirable item – We have added the following information: “The colony was established from ~1000 adults generated from four weekly shipments of ~2000 eggs each. Eggs were reared to adulthood following Dr. Wade French’s protocol [10].”
L157-158: the desired number of eggs – please give the thresholds (10-100 etc). We have added the requested information to the text.
L160: mold growth – where exactly? Filter paper and eggs. We have added the requested information to the text.
L168: small holes – diameter is welcome. We have added the requested information to the text.
Ibidem: water – can some approximate amount (volume ratio) be given? Unfortunately, we do not have accurate data for this due to variability of moisture levels in purchased bags of soil. Hence the use of “pinching” soil between our fingers to determine need for additional water.
L82: days-AEL – hyphen seems to be odd here. Thanks for catching this. The hyphen has been removed.
L206: desiccating has been changed to “avoid desiccation”.
L225: data to determine – what kind of data is meant? developmental and survivorship which has been added to the text.
L233: still looked like larvae – sounds like there are insects which are not larvae but look like larvae :) please rephrase. Thanks for catching this! We, edited the sentence to read: “with third-instar larvae being the largest individuals still moving and feeding, and pre-pupae being those that had already built a soil cocoon or were otherwise showing little movement.”
L237-238: to set up a quality control system for monitoring the rearing system = to set up a quality control for monitoring the rearing system (avoid excessing word repetition, “control” in this case) Thanks for catching this! We changed it to: “A full year’s worth of recorded survival rates was used to set up a quality control for monitoring the rearing system.”
L238-238 from the third-instar larval stage to the adult stage = from the third-instar larva to the adult. This has been changed.
L254: rates calculated = rates were calculated. This has been changed.
L277-278: “pairs vs chambers” doesn’t give a clear comprehension if there were ten pairs per chamber or ten pairs per ten chambers. Thanks for catching this! We changed it to: “A single-pair of WCR adults were placed into each of ten mating chambers (see 2.2.3)…”.
L279: quantify egg-laying – consider “quantify the egg-laying”. We changed this to: “quantify the egg-lay”.
L281: he was – consider “it was”. This has been changed.
L284: track parentage – consider “track the parentage”. This has been changed.
L300: lab we – consider comma usage. This has been changed.
L308-310: please build up this conclusion upon some previously published data. Thanks for pointing this out. We changed the section from:
“More than 80% of the adults eclosed between 41 and 45 days (Figure 6). The short, high peak seen for adult emergence indicates that the system was stable, and that the insects developed at similar rates.”
To:
“More than 80% of the adults eclosed between 41 and 45 days (Figure 6), which is similar to other WCR rearing protocols [5], both for overall survival rates from larvae to adult, as well as development time. Thus, we conclude that the short, high peak seen for adult emergence indicates that the system was stable, and that the insects developed at similar rates.”
L326: you may want to mention the overall period of observation spanned by the mean survival rate. To address this, we the following to the end of that paragraph: “It should be noted that our survival data (Figure 7) indicate the rearing system was more stable the second half of the year. This suggests that when setting up a new rearing system it might take a few generations (in this case 3 generations) to stabilize the rearing system. However, this data can be helpful for generating a robust quality control system.”
Table 2 – the CFP type is indicated in the heading and is homogenous for all variants, the first (from the left) column should be removed. Thanks for catching this! The first column has been deleted.
L473: what does the sentence in the parentheses mean? The (< 1 box per month vs. 40 boxes per week) was intended to explain that by controlling for the number of plants per container, mite and mold issues were reduced from being found in 40 rearing containers per week, to only 1 per month. To help clarify we changed to “Specifically, the occurrence of mites and mold was reduced forty-fold, from ~40 boxes per week to less than 1 box per month”.
L482: do you consider the work performed by workers of three different levels as independent replicates and compare the data between to access reproducibility? No, these were not meant to be replicates so we have added clarification of that point to the text.
L556-567: this last paragraph may be headed as “Conclusion” as a lot of information herein is a repetition of ideas already given in Discussion. Excellent point! We have made the suggested change.